# The Intranigral Infusion of Human-Alpha Synuclein Oligomers Induces a Cognitive Impairment in Rats Associated with Changes in Neuronal Firing and Neuroinflammation in the Anterior Cingulate Cortex

**DOI:** 10.3390/cells11172628

**Published:** 2022-08-24

**Authors:** Maria Francesca Palmas, Michela Etzi, Augusta Pisanu, Chiara Camoglio, Claudia Sagheddu, Michele Santoni, Maria Francesca Manchinu, Mauro Pala, Giuliana Fusco, Alfonso De Simone, Luca Picci, Giovanna Mulas, Saturnino Spiga, Maria Scherma, Paola Fadda, Marco Pistis, Nicola Simola, Ezio Carboni, Anna R. Carta

**Affiliations:** 1Department of Biomedical Sciences, University of Cagliari, 09040 Cagliari, Italy; 2National Research Council, Institute of Neuroscience, 09040 Cagliari, Italy; 3Istituto Di Ricerca Genetica e Biomedica Del Consiglio Nazionale Delle Ricerche, 09040 Monserrato, Italy; 4Centre for Misfolding Diseases, Department of Chemistry, University of Cambridge, Cambridge CB2 1EW, UK; 5Department of Pharmacy, University of Naples “Federico II”, 80131 Naples, Italy; 6Department of Life and Environmental Sciences, University of Cagliari, 09040 Cagliari, Italy

**Keywords:** Parkinson’s disease, cognitive impairment, neuroinflammation, α-synuclein, neuronal activity, microglia

## Abstract

Parkinson’s disease (PD) is a complex pathology causing a plethora of non-motor symptoms besides classical motor impairments, including cognitive disturbances. Recent studies in the PD human brain have reported microgliosis in limbic and neocortical structures, suggesting a role for neuroinflammation in the development of cognitive decline. Yet, the mechanism underlying the cognitive pathology is under investigated, mainly for the lack of a valid preclinical neuropathological model reproducing the disease’s motor and non-motor aspects. Here, we show that the bilateral intracerebral infusion of pre-formed human alpha synuclein oligomers (H-αSynOs) within the substantia nigra pars compacta (SNpc) offers a valid model for studying the cognitive symptoms of PD, which adds to the classical motor aspects previously described in the same model. Indeed, H-αSynOs-infused rats displayed memory deficits in the two-trial recognition task in a Y maze and the novel object recognition (NOR) test performed three months after the oligomer infusion. In the anterior cingulate cortex (ACC) of H-αSynOs-infused rats the in vivo electrophysiological activity was altered and the expression of the neuron-specific immediate early gene (IEG) *Npas4* (Neuronal PAS domain protein 4) and the AMPA receptor subunit GluR1 were decreased. The histological analysis of the brain of cognitively impaired rats showed a neuroinflammatory response in cognition-related regions such as the ACC and discrete subareas of the hippocampus, in the absence of any evident neuronal loss, supporting a role of neuroinflammation in cognitive decline. We found an increased GFAP reactivity and the acquisition of a proinflammatory phenotype by microglia, as indicated by the increased levels of microglial Tumor Necrosis Factor alpha (TNF-α) as compared to vehicle-infused rats. Moreover, diffused deposits of phospho-alpha synuclein (p-αSyn) and Lewy neurite-like aggregates were found in the SNpc and striatum, suggesting the spreading of toxic protein within anatomically interconnected areas. Altogether, we present a neuropathological rat model of PD that is relevant for the study of cognitive dysfunction featuring the disease. The intranigral infusion of toxic oligomeric species of alpha-synuclein (α-Syn) induced spreading and neuroinflammation in distant cognition-relevant regions, which may drive the altered neuronal activity underlying cognitive deficits.

## 1. Introduction

Non-motor symptoms greatly contribute to Parkinson’s disease (PD) burden in addition to cardinal motor impairment [1]. Cognitive decline is among the most recurrent symptoms that occur during disease progression in PD, often slily starting in early stages as mild cognitive impairment (MCI) and evolving later to dementia [2,3,4]. The cognitive dysfunction in PD encompasses a range of deficits, mostly exhibited as reduced memory, executive function, attention and visuospatial domains [5], with almost half of patients developing PD dementia (PDD) within 10 years from diagnosis [6].

In front of the recognition of MCI as a recurrent non-motor symptom in PD, the neuropathological basis is still largely unclear. The pathological role of alpha-synuclein (α-Syn) in cognitive impairment has been suggested by studies of elderly PD patients in advanced PD stage. In these patients, the α-Syn pathology in limbic brain regions, such as the entorhinal and anterior cingulate cortex (ACC), the hippocampus and the amygdala, correlated with cognitive decline, suggesting that Lewy Bodies (LB) in these areas may predict the development of PDD [7,8,9,10]. A significantly higher burden of LB-pathology was described in PDD as compared with PD without dementia, across multiple brain regions [11]. However, the MCI is common in PD patients in early disease phases or even in newly diagnosed patients, when neocortical LB deposits are not yet detectable [2,12,13], suggesting that the neuropathology beneath cognitive decline initiates before a clearly detectable protein deposition can be described, and leaving uncertain the role of α-Syn in MCI.

Besides αSyn pathology, neuroinflammation is a hallmark of PD. However, despite the clear contribution to neurodegeneration, the role of dysfunctional glial cells in the neuropathology of cognitive symptoms in PD has been poorly investigated [14]. Positron Emission Tomography (PET) neuroimaging studies using the microglial ligand [11C] PK11195 have suggested that reactive microglia are increased in PDD [15]. Increased expression of the microglial markers CD68 and MHC-II has been described in limbic brain regions of PD patients, although this feature was not investigated in relation to cognitive impairment [7,16]. Moreover, reactive astroglia was observed in cognitive areas, including the ACC, in early PD [17]. Interestingly, HLA-DR^+^ microglia described in some limbic regions of PDD patients did not consistently overlap the LB pathology [8], and markers of reactive microglia are not a feature of dementia with LB [18,19]. Finally, an increase of inflammatory monocytes and expression of their toll-like receptors (TLR)-2 and TLR-4 has been correlated with a high risk of dementia in PD patients [20]. Surprisingly, although soluble factors including cytokines are the effector molecules for a multitude of glial functions, only one single study investigated cytokine changes in relation to cognitive symptoms in PD, showing a more pronounced increase of pro-inflammatory cytokines in limbic regions of PDD patients [8]. Through the release of cell signaling factors such as cytokines, microglia play a pivotal neuromodulatory role, influencing synapse connectivity and remodeling, as well as neuronal membrane properties, being heavily involved in the brain homeostasis and likely in cognitive processes and neuropsychiatric disorders [21,22,23]. In PD, chronically dysfunctional microglia have been suggested to profoundly affect synaptic connectivity in motor areas through the excessive release of inflammatory cytokines [24,25]. Importantly, the pro-inflammatory cytokine Tumor Necrosis Factor alpha (TNF-α) is a key regulator of neuronal excitability, synaptic strength and plasticity via the modulation of several target molecules such as the α-amino-3-hydroxy-5-methyl-4-isoxazolepropionic acid (AMPA) glutamate receptor subunit 1 (GluR1) [24,26].

A main reason for the poor knowledge of MCI neuropathology is the lack of a preclinical neuropathological model that may reproduce both the classical motor dysfunction and the non-motor cognitive symptoms. In recent years, a novel translational model of PD neuropathology has been developed, based on the intracerebral infusion of in vitro-generated human α-Syn oligomers (H-αSynOs) [27,28,29], according to the evidence that soluble small oligomeric species are the most toxic form of this protein [30,31,32]. The H-αSynOs model progressively develops motor deficits, and displays a typical α-Syn-induced neuropathology, including the spreading and aggregates of phosphorylated α-Syn (p-αSyn) and a chronic unremitting neuroinflammation in basal ganglia structures primarily involved in the motor pathology, along with a systemic inflammatory status [27,29].

Here, we show that the bilateral intranigral infusion of H-αSynOs-induced memory deficits is characterized by the loss of short-term spatial reference memory and recognition memory when measured three months after infusion. The memory impairment was paralleled by an impaired neuronal activity in the ACC. Moreover, we found that the H-αSynOs infusion induced a neuroinflammatory response in cognition-relevant areas such as the ACC and hippocampus. In these regions, microglia underwent poor proliferation but acquired a proinflammatory phenotype recognized by increased TNF-α production. Spreading of p129-αSyn was indicated by the presence of large protein deposits within the infused substantia nigra (SN) as well as the striatum, suggesting that pathological α-Syn spread through anatomically interconnected areas. Taken together, our results show that the intranigral infusion of toxic oligomeric species of α-Syn induced memory impairment in rats underlaid by a neuroinflammatory environment in distant cognition-relevant regions, which may drive the altered neuronal activity and cognitive deficits.

## 2. Materials and Methods

### 2.1. Expression and Purification of Recombinant Human αSyn (H-αSyn)

Plasmid pT7-7 encoding for the protein was used to express and purify H-αSyn in E. coli, as described [27]. Briefly, BL21 (DE3)-gold cells (Agilent Technologies, Santa Clara, CA USA) were used for transforming H-αSyn. After that, H-αSyn was obtained by growing the bacteria at 37 °C under constant conditions [27] and protein expression was induced with 1 mM isopropyl β-D-1-thiogalactopyranoside (IPTG) at 37 °C for 4 h. The cells were then harvested by centrifugation at 6200× *g* (Beckman Coulter, Brea, CA USA). After pellet resuspension in lysis buffer [27] and sonication, the cell lysate was centrifuged at 22,000× *g* for 30 min, the supernatant was heated for 20 min at 70 °C and centrifuged again at 22,000× *g*. Streptomycin sulfate was added to the supernatant [27] and the mixture was stirred for 15 min at 4 °C followed by further centrifugation at 22,000× *g*. The protein was precipitated with ammonium sulfate to a concentration of 360 mg·ml^−1^ and the solution was stirred for 30 min at 4 °C and centrifuged again at 22,000× *g*. The resulting pellet was resuspended, dialyzed to remove salts [27] and loaded onto an anion exchange column (26/10 Q Sepharose high performance, GE Healthcare, Little Chalfont, UK). Eluate was obtained with a 0–1 M NaCl step gradient, and then further purified by loading onto a size exclusion column (Hiload 26/60 Superdex 75 preparation grade, GE Healthcare, Little Chalfont, UK). Fractions containing the monomeric protein were pooled together and concentrated by Vivaspin filter devices (Sartorius Stedim Biotech, Gottingen, Germany). Protein purity was SDS-PAGE-analyzed, and concentration was determined spectrophotometrically using 275 = 5600 M^−1^ cm^−1^.

### 2.2. Purification of H-αSyn Oligomers 

Toxic oligomeric samples of recombinant H-αSyn were prepared as previously described [30,31]. Briefly, 6 mg of lyophilized protein was resuspended in PBS buffer, the solution was passed through a 0.22 μm cut off filter and subsequently incubated at 37 °C for 24 h in stationary mode and without agitation to minimize fibril formation. Residual fibrils were removed by ultracentrifugation for 1 h at 288,000× *g* using a TLA-120.2 Beckman rotor (Beckman Coulter, Brea, CA, USA). The excess of αSyn monomers was removed from the sample by multiple filtration steps using 100 kDa cutoff membranes, resulting in the enrichment of the H-αSynOs species. H-αSynOs prepared with this protocol have been found stable for several days, and in this study were used within one week of their production. 

### 2.3. Animals and Stereotaxic Surgery

12-weeks old male Sprague Dawley rats (275–300 g) were purchased from Envigo. All rats were housed in groups of four in polypropylene cages, with food and water available *ad libitum*, in rooms maintained at 21 °C under a 12 h light/dark cycle (lights on 7:00 A.M.). 34 rats were deeply anesthetized with Fentanyl (0.33 mg/kg, i.p.) and medetomidine hydrochloride (0.33 mg/kg, i.p.), 5 µL of H-αSynOs were stereotaxically injected bilaterally into the substantia nigra pars compacta (SNpc) at the rate of 1 µL/min via a silica microinjector, as previously described (coordinates relative to bregma; −5.4 mm anteroposterior; ±1.9 mm from the midline; −7.2 mm beneath the dura, according to the atlas of Paxinos and Watson) [27,29,33]. Control animals received an equal volume of sterile phosphate buffer saline (PBS, pH 7.4) at the same infusion rate (Figure 1). All procedures were performed in accordance with the ARRIVE guidelines and with the guidelines approved by the European Community (2010/63UE L 276 20/10/2010). Protocols were approved by the Italian Ministry of Health (authorization n° 766/2020-PR). All efforts were made to minimize pain and discomfort and to reduce the number of experimental animals used.

### 2.4. Behavioral Tests

Short-term memory performance and locomotor activity were assessed over a 7-day interval, beginning 12 weeks after surgery. Prior to behavioral testing, animals were brought into the procedure room and allowed to habituate to the room for 30 min in order to avoid any alterations in behavioral parameters induced by the novel environment. Tests were carried out during the light phase (9:00–15:00 h). All tests were performed and analyzed in a blinded fashion.

#### 2.4.1. Two-Trial Recognition Test in a Y Maze

A two-trial recognition test in a Y-maze was performed to assess the effects of H-αSynOs infusion on spatial recognition memory. The Y maze used was made of black PVC and had three equally sized symmetrical arms (L 50 cm × W 20 cm × H 35 cm) that converged onto a central triangular area. The arms were randomly designated as: “start arm”, in which the rat always started to explore the maze, “novel arm”, and “other arm”. Testing was performed by individually placing each rat in the “start arm” of the maze and involved two trials separated by a 1 h interval [34]. The novel arm was blocked by a guillotine door during trial 1 and left open during trial 2, whilst the start and the other arms were left open during both trial 1 and 2. The bottom of the maze was covered with sawdust that was replaced after testing each rat in order to remove olfactory cues. The maze was placed in a quiet room with dim illumination, and visual cues were placed outside the walls of the maze, in order to allow maze navigation by rats.

During Trial 1 (10 min), each rat was left free to explore only the “start” and the “other” arm, with access to the “novel” arm being blocked. During Trial 2 (5 min), each rat was left free to explore all the three arms of the maze (“novel”, “start”, “other”). Rats’ performance was videotaped and later analyzed to score the number of entries in each arm and the amount of time (seconds) spent in each arm. Spatial recognition memory was assessed by evaluating the preference for the “novel” arm vs the combination of “start” and “other” arms, expressed as number of seconds spent and entries performed in the arms during the second trial [35].

#### 2.4.2. Novel Object Recognition Test

Novel Object Recognition (NOR) test was carried out in a black box (60 × 60 cm) according to a previously published protocol [36]. After the habituation session (10 min, T_0_), rats were re-placed individually in the test box containing two identical objects for 10 min, before returning them into their home cage (familiarization phase, T_1_). Sixty minutes later, rats were placed again for 5 min in the same test chamber containing one familiar and one novel object (choice phase, T_2_). Training and test sessions were recorded with a camera. Object recognition was expressed by the discrimination index (DI) according to the following formula: (Tn − Tf)/(Tn + Tf) (Tn = time spent exploring the novel object; Tf = time spent exploring the familiar one).

### 2.5. Immunohistochemistry

After the behavioral testing, rats were deeply anesthetized and transcardially perfused in ice-cold 0.1 M PBS (pH 7.4) followed by 4% buffered paraformaldehyde. After the perfusion, brains were carefully removed from the skull, post-fixed overnight in 4% paraformaldehyde-PBS and stored in 0.1% NaN3-PBS at 4 °C. Serial coronal sections of ACC, dorsal hippocampus and midbrain were vibratome-cut (40 µm thickness).

For p129-αSyn visualization, sections from midbrain, hippocampus and ACC were placed into 24-well plates and pre-incubated in normal donkey serum and then immunoreacted with rabbit monoclonal anti p129-αSyn (1:800, Abcam, Cambridge, UK) primary antibody. The proper biotinylated secondary antibody was used to amplify the reaction and the classic avidin-peroxidase complex was used for visualization (ABC, Vector, UK), using the chromogen 3,30-diaminobenzidine (Sigma-Aldrich, St. Louis, MO, USA). 

For immunofluorescence, after pre-incubation of hippocampal and ACC sections in a blocking solution, the following unconjugated primary antibodies were used for single or double immunolabeling: goat polyclonal anti Iba-1 (1:1000; Novus Biologicals, Littleton, CO, USA); rabbit polyclonal anti TNF-α (1:500, Novus Biologicals, Littleton, CO, USA); rabbit polyclonal anti GFAP (glial fibrillary acidic protein, 1:2500, Novus Biologicals, Littleton, CO, USA); mouse monoclonal anti GluR1 (1:200, Novus Biologicals, Littleton, CO, USA). For fluorescence visualization of Iba-1, GluR1 and GFAP a two-step indirect labelling protocol was used; in order to increase the signal of TNF-α a three-step detection was performed by combining biotin-SP-conjugated IgG (1:500, Jackson Immunoresearch, West Grove, PA, USA) and streptavidin–fluorescein (1:400, Jackson Immunoresearch, West Grove, PA, USA), as described [37]. Images were acquired by a spinning disk confocal microscope (Crisel Instruments, Rome, Italy) with a ×63 magnification.

### 2.6. Cresyl Violet Staining

Cresyl violet (CV) staining was carried out in order to qualitatively examine the neuronal survival in the hippocampus and ACC. Briefly, once mounted onto microscopy slides, the brain sections were stained with a solution of 0.1% *w*/*v* CV-acetate (ThermoFisher Scientific, Waltham, MA, USA) and then dehydrated in serial ethanol bath.

### 2.7. Fluorescence Microscopy Analysis 

A spinning disk confocal microscope (Crisel Instruments, Rome, Italy) was used for qualitative and quantitative analyses of Iba-1, TNF-α, GluR1 and GFAP at ×63 magnification. Surface rendering, colocalization, maximum intensity and simulated fluorescence process algorithms were applied (ImageJ and Imaris 7.3) before obtaining a stack from each dataset (40 images) to determine the Iba-1, GluR1 and GFAP occupied volume. In the resulting stacks, 10 regions of interest for the ACC, granular cell layer (GCL) and hilus of dentate gyrus (DG), CA3 pyramidal layer (x = 700 μm; y = 700 μm; z = 40 μm) and 5 regions of interest for the CA1 subfields (x = 1024 μm; y = 1024 μm; z = 40 μm) in each acquired section and for each animal were randomly chosen, and the volume of the elements calculated. The GFAP analysis was conducted only at the ACC level. For colocalization analysis, a colocalization channel was automatically generated by Imaris 7.3. Colocalized TNF-α was analyzed in the same sections and ACC/hippocampal regions were Iba-1 was analyzed (see above). A stack was obtained from each dataset (40 images). In the resulting stacks, each Iba-1^+^ cell was identified and selected, and the volume of the colocalized TNF-α was calculated within each cell (μm^3^). Specifically, three dimensional isosurfaces were created with the Surfaces module from both Iba-1 and TNF- α channels. A new channel was then generated from the colocalization, and a 3D isososurface of overlapping areas was created.

### 2.8. In Vivo Single Unit Recordings from the ACC 

Three months after the bilateral surgery in the SNpc, rats were deeply anesthetized and placed in a stereotaxic apparatus. Single-unit activity of putative pyramidal neurons from the ACC (AP: +1.0–1.6 mm from bregma, L: 0.3–0.8 mm from the midline V: 1.5–4.0 mm from the cortical surface) was recorded with glass micropipettes filled with 2% Pontamine sky blue dissolved in 0.5 M sodium acetate. Individual action potentials were isolated and amplified (bandpass filter 10–5000 Hz) by means of a Neurolog system (Digitimer, Hertfordshire, UK) or a CP511 AC Amplifier (Grass Instruments Co., West Warwick, RI, USA). Experiments were sampled with Spike2 software by a computer connected to a CED1401 interface (Cambridge Electronic Design, Cambridge, UK). Cells were selected in accordance with electrophysiological characteristics attributed to pyramidal neurons [38,39,40] which present “regular-spiking” or “intrinsically bursting” activity of biphasic and >2 ms-wide action potentials.

### 2.9. RNA Isolation, Library Preparation and Sequencing

RNA was extracted from two brain regions (ACC and HP) using the PureLink^®^ RNA Mini Kit (Ambion #12183018A) according to the manufacturer’s instructions. The quantity and the integrity of isolated RNA samples were evaluated using the Agilent 2100 Bioanalyzer (Agilent Technologies, Palo Alto, CA, USA). RNA-seq libraries were prepared with an Illumina^®^ Stranded mRNA Prep, Ligation Kit (Illumina #20040532) and IDT^®^ for Illumina^®^ RNA UD Indexes Set A, Ligation (Illumina #20040553). The RNA library concentration was measured using a Qubit^®^ 2.0 fluorometer and successfully sequenced on an Illumina NovaSeq6000 (Illumina Inc., San Diego, CA, USA).

### 2.10. RNAseq Data Analysis

The RNAseq short reads quality was evaluated with FASTQC (v0.11.9) [41] software. The rattus norvegicus reference genome (primary assembly, mRatBN7.2) was downloaded from the ENSEMBL web site (http://ftp.ensembl.org (accessed on 18 June 2021)). Short reads were aligned to the reference genome with STAR software (2.7.9a) [42]. The genes coordinates (Rattus_norvegicus.mRatBN7.2.105.gtf) were downloaded from the ENSEMBL web site (http://ftp.ensembl.org (accessed on 18 June 2021)) and the gene expression level was evaluated with HTSeq software [43] (0.11.3) with the following command line: htseq-count − stranded = reverse − mode = union − idattr = gene_id − type = exon. Potential latent confounders were inferred with the svaseq software [44].

Only genes that had at least 5 reads in at least the 25% of the samples were analyzed for differential expression (between treated and untreated) with the DESeq2 software [45] by using defaults settings. We performed differential expression both with and without incorporating the svaseq confounders in the DESeq2 model. The FDR was computed with the Benjamini-Hochberg method.

### 2.11. Reverse Transcription-Quantitative PCR (RT-qPCR)

Total RNA was extracted using PureLink^®^ RNA Mini Kit as described above. The cDNA was synthetized using random primers and a reverse transcription kit SuperScript™ III First-Strand Synthesis System (Cat# 18080-051, Invitrogen) according to the manufacturer’s procedure. RT-qPCR was performed using Platinum SYBR^®^ Green (Cat # 11744-100 Invitrogen) on ABI PRISM 7900 thermocycler (Applied Biosystems, Foster City, CA, USA). The following primers were used:*Gapdh*: Fw 5′GGCTGCCTTCTCTTGTGACA 3′-Rev 5′ TGAACTTGCCGTGGGTAGAG 3′*β-Actin*: Fw 5′ TCAACACCCCAGCCATGTAC 3′-Rev 5′ TCCGGAGTCCATCACAATGC 3′*Npas4*: Fw 5′ ATCAGTGACACGGAAGCCTG 3′-Rev 5′ AGCTGGGGTTCCTAGGACAT 3′*Npas4*: Fw 5′ GATCGCCTTTTCCGTTGTCG3′-Rev 5′ CAGGTGGGTGAGCATGGAAT 3′.

The target gene expression level was normalized to *Gapdh* and *β-actin* mRNA expression levels. RT-qPCR was performed in triplicate and the analysis data was done using the ΔΔCT method.

### 2.12. Statistical Analysis 

Researchers blinded to experimental conditions evaluated the outcome measures. Results are presented as mean ± SEM (Statistica 8, Stat Soft Inc., Tulsa, OK, USA). Normal distribution was assessed by Kolmogorov–Smirnov test in all data sets and for all experimental designs. Where normality was respected, data were analyzed by two-way ANOVA followed by Tukey’s post-hoc test and unpaired Student’s *t* test. Where not, data were analyzed by the non-parametric Mann–Whitney test. The level of significance was set at *p* < 0.05.

## 3. Results

### 3.1. The Intranigral H-αSynOs Infusion Induced a Cognitive Impairment

Rats infused with H-αSynOs within the SNpc displayed a memory impairment as revealed by both the two-trial recognition test in a Y maze and the NOR. Specifically, rats displayed a reduced preference for the novel arm in the two-trial recognition test. Two-way ANOVA for seconds spent in arms revealed a significant effect of arm (F_1,36_ = 10.35; *p* = 0.027), but neither an effect of infusion (F_1,36_ = 1.75; *p* = 0.19) nor an interaction arm × infusion (F_1,36_ = 2.67; *p* = 0.11) (Figure 2A). Tukey’s post-hoc test showed that rats infused with vehicle (n = 10) spent significantly more time in the novel arm of the maze than in the remaining two arms of the maze (here expressed as the mean value between the time spent in the ‘entry’ arm and in the ‘other’ arm) (*p* < 0.01). Conversely, rats infused with H-αSynOs (n = 10) spent a similar amount of time in the arms, as revealed by the lack of any significant effect of arm (F_1,36_ = 0.58; *p* = 0.45) and infusion (F_1,36_ = 0.09; *p* = 0.76) and no significant interaction of arm × infusion (F_1,36_ = 0.24; *p* = 0.63) (Figure 2B).

Results from the NOR test revealed a significantly lower recognition index in H-αSynOs-treated rats (n = 10) compared to the vehicle-infused rats (n = 10) (t_(18)_ = 3.19; *p* = 0.005 by Unpaired Student’s *t*-test) in the test phase (Figure 2C). No differences were seen in the preference for location between the two groups during the training session (data not shown).

In the same rats, we confirmed the presence of motor impairment shown in our previous studies in the same PD model, by means of the challenging beam walk test (Appendix A) [27].

### 3.2. H-αsynOs Infusion Altered Neuronal Activity in the ACC 

We performed single-unit recordings of putative pyramidal neurons within the ACC to assess the *H-αsynOs* infusion effects. The number of spontaneously active pyramidal neurons was significantly reduced in H-αSynOs-infused rats (n = 6) as compared with sham-operated rats (n = 5) (*p* = 0.026 by Unpaired Student’s *t* test) (Figure 3B). The mean firing rate was also reduced (Figure 3C; *p*= 0.024; Unpaired Student’s *t* test) in H-αSynOs-infused rats (n_(treated)_ = 33) as compared with sham-operated controls (n_(sham)_ = 39). The mean coefficient of variation among interspike intervals recorded from pyramidal neurons did not differ between experimental group (n = 33–39; *p* > 0.05 by Unpaired Student’s *t* test) (data not shown).

### 3.3. Npas4 Expression was Downregulated after H-αsynOs Infusion in Both the ACC and Hippocampus

The transcriptional profile of H-αSynOs-infused rats was compared with vehicle-infused animals. We performed Bulk RNA sequencing in the dissected ACC and hippocampus of H-αSynOs-infused rats (n = 6) and compared it with vehicle-infused rats (n = 6). We identified 484 (ACC) and 321 (hippocampus) differentially expressed genes with nominal *p* value < 0.05 after correcting for potential latent confounders across the experimental groups in both areas [46]. Among differently expressed genes, we focused on the neuron-specific immediate-early gene (IEG) *Npas4* (Neuronal PAS domain protein 4), whose expression was downregulated in both the ACC (log2FoldChange = − 7.52 × 10^−6^; *p* < 0.005) and hippocampus (log2FoldChange = − 0.7692225; *p* < 0.001) of H-αSynOs-infused rats as compared to the vehicle counterpart.

RT-qPCR analysis in the same tissues confirmed the H-αSynOs-induced inhibition of *Npas4* mRNA expression in the ACC (*p* = 0.0005; Unpaired Student’s *t* test) (Figure 3D) and in the hippocampus (*p* < 0.0001; Unpaired Student’s *t* test) (Figure 3E), indicating that the memory impairment was sustained by an altered neuronal activity in cognition-related areas.

### 3.4. The Cognitive Impairment Induced by H-αSynOs Infusion was Associated with a Dysregulated Glial Activity in ACC and Hippocampus

#### 3.4.1. Iba-I Immunofluorescence

Iba-1 IF was calculated as the total volume occupied by Iba-1^+^ cells in the aforementioned limbic areas. Moreover, due to the high morphological heterogeneity of microglial cells within the same neuroanatomical area, single subfields were analyzed. As shown in Table 1, no differences were seen in the ACC (both in the III and V layer) between H-αSynOs-infused rats and sham-operated rats. In the dorsal hippocampus, Mann–Whitney non-parametric test revealed an increase of Iba-1^+^ cell volume only in the pyramidal layer of CA1 subfield (*p* = 0.0023) and in the hilus of the DG (*p* = 0.0335) of H-αSynOs-infused rats (Table 1).

#### 3.4.2. TNF-α Immunofluorescence

TNF-α was specifically analyzed as a master regulator of the inflammatory cascade and for its important role as a regulator of neuronal excitability. Colocalization analysis was performed for TNF-α within microglial cells in the same areas and subfields analyzed for Iba-1 IF. As shown in Figure 4 and Figure 5, Iba-1+ cells of H-αSynOs-infused rats displayed a substantial increase of TNF-α levels both in the ACC and the hippocampus. Specifically, the TNF-α increase was evident in both the cortical layers III and V (Figure 4; *p*_(III layer)_ < 0.001; *p*_(V layer)_ = 0.005 by Mann–Whitney non-parametric test) and in the pyramidal layer of CA1 (*p* = 0.0238; Mann–Whitney non-parametric test) and CA3 (*p* < 0.0001; Mann–Whitney-non parametric test) (Figure 5).

#### 3.4.3. GFAP Immunofluorescence

Given the critical role of astroglial cells in neuroinflammatory responses, as well as in the regulation of neuronal activity, GFAP IF was analyzed in the ACC as an astrocyte marker. Specifically, the total volume occupied by GFAP+ cells was calculated in the ACC layers III and V, showing an increase of GFAP IF in H-αSynOs-infused rats as compared to vehicle-infused rats (Figure 6) (*p*_(III layer)_ = 0.0378; *p*_(V layer)_ = 0.0152 by Mann–Whitney non-parametric test).

Importantly, neuroinflammation in these limbic regions was not associated with neuronal loss, suggesting that glial activation did not sustain neurodegenerative mechanisms, and the cognitive decline was not underlaid by a neurodegenerative process (Figure 7).

#### 3.4.4. GluR1 Immunofluorescence

Changes in GluR1 protein levels were investigated by IF in the ACC as a neuronal marker of synaptic activity and because it is a recognized target molecule of TNF-α actions. The confocal microscopy analysis revealed a significant reduction in GluR1 levels in the ACC of H-αSynOs-infused rats as compared with vehicle-infused rats (Figure 8) (*p* = 0.0057 by Mann–Whitney non-parametric test).

### 3.5. P129-αSyn Aggregates were Detected along the Nigrostriatal Pathway

Based on previous evidence of αSyn spreading throughout the connectome, we evaluated the presence of aggregated forms of p129-αSyn in the infusion site as well as in the projection area nucleus striatum. Moreover, we sought p129-αSyn in the same cortical regions evaluated for inflammatory responses. Immunohistochemistry showed round-shaped deposits of p129-αSyn in the SNpc and diffused neurite-like deposits in the whole striatum of H-αSynOs-infused-rats (Figure 9). In the ACC and the hippocampus, besides very small immunopositive dots, we did not detect clear evident aggregates of p129-αSyn in these regions (data not shown).

## 4. Discussion

The present study demonstrates a causal link between H-αSynOs-induced toxicity and memory decline in an experimental model of PD, and provides evidence that neuroinflammation may act as the trait d’union of αSyn-induced neuropathology and impaired cognition. In synopsis, we show that the intranigral infusion of H-αSynOs was sufficient to induce a memory decline in rats, functionally underlined by altered neuronal activity in the ACC and hippocampus. Moreover, the memory decline was associated with increased microglial production of the inflammatory cytokine TNF-α in the same cognition-related regions. Astrocytosis was also found in the same areas. Large deposits of p-αSyn were detected in the infused SN, as well as in the whole striatum of H-αSynOs-infused rats, but not in the analyzed cognition-related regions.

### 4.1. The Intranigral Infusion of H-αSynOs Induced a Cognitive Decline and Suppressed Cortical Activity

Amongst non-motor disturbances that may affect PD patients, cognitive dysfunction is highly recurrent, mostly exhibited as reduced memory, executive function, attention and visuospatial domains [3,4,47]. Despite the large incidence of cognitive decline in the PD-affected population, its neuropathological substrate is largely unknown. A prominent role of LB pathology has been suggested in PDD mainly by studies in PD patients [8,11], however, the neuropathological substrate of MCI, which occurs in early PD stages and can be considered the prodromal phase of PDD, is unclear and has been poorly investigated preclinically [2]. Here, we used the PD model obtained with the intranigral H-αSynOs infusion, based on solid evidence suggesting that the small soluble prefibrillar oligomers are the most toxic species of αSyn, which have been described in selected areas of the PD brain, as well as in the biological fluids of PD patients [30,48,49,50,51,52,53]. We exploited a validated method to prepare and purify αSyn oligomers that are relatively homogeneous in size and structure [31,32]. We demonstrated that oligomeric αSyn can act as a first trigger in the development of memory decline by showing that the bilateral H-αSynOs intranigral infusion induced an impairment in the execution of behavioral tests specifically aimed at evaluating short-term spatial reference memory and recognition memory when rats were tested three months post-infusion. The in vivo protocol used in the present study is based on a previously validated PD model, where the unilateral H-αSynOs infusion reproduced the motor symptomatology and the underlying progressive neuropathology, including dopaminergic degeneration, persistent neuroinflammation in motor-related areas as well as systemic immune dysregulation, deposition and spreading of p-αSyn outside the infusion site [27]. Moreover, the predictive validity as a translational PD model to test disease-modifying compounds has been recently reported [29]. Here, a bilateral infusion protocol was adopted in order to avoid any contralateral compensation in cognition tasks. We performed the two-trial recognition task in a Y maze to evaluate the short-term spatial reference memory and found that H-αSynOs-infused rats were unable to discriminate between the arms of the maze, since they spent a comparable amount of time in novel and familiar arms [35]. Importantly, the number of entries in the arms did not differ across experimental groups, indicating that spatial memory impairment was not attributable to the presence of motor deficits. The impairment in short term-memory was further demonstrated by the NOR, where H-αSynOs-infused rats were unable to discriminate between novel and familiar objects. Therefore, the present results extend the characterization of this PD model, indicating that the intranigral H-αSynOs infusion reproduces both motor and non-motor cognitive aspects of the pathology.

The memory impairment was functionally supported by an impaired neuronal activity measured in the ACC. Among limbic brain regions, dysfunction of the ACC, as well as the hippocampus, seem to heavily contribute to cognitive decline in PD [54,55,56,57,58,59,60]. In our study, in vivo extracellular recording showed a reduction of spontaneous neuronal firing in the ACC of rats infused with H-αSynOs. In addition, the expression of *Npas4* was downregulated in the same area, as well as in the hippocampus, of H-αSynOs-infused rats. *Npas4* is an IEG involved in the activity-dependent regulation of the excitatory-inhibitory balance, and it is of high relevance for the present study because it represents an important molecular link between neuronal activity and memory. Moreover, *Npas4* is expressed only in neurons, is among the most rapidly induced IEGs, and is selectively induced by neuronal activity [61,62]. Therefore, both the electrophysiological and molecular evidence indicate a reduction of neuronal activity, which may be a functional correlate of the deficits in cognitive performance observed in behavioral studies. 

Cortical and limbic LB deposition has been described in the brain of PDD patients and is considered to correlate with cognitive decline in PD [8]. However, other studies have reported that MCI may already develop in the early or even prodromal stages of the disease, when no or few cortical LB are present [12], arguing against a clear-cut correlation between LB pathology and cognitive impairment and suggesting that other under-explored mechanisms may contribute to cognitive decline in PD [6]. In our study, we detected prominent p-αSyn deposits and neurite-like aggregates in the infused SN and in the nucleus striatum, confirming that p-αSyn spread through anatomically interconnected areas after the oligomer infusion. Moreover, we did not detect p129-αSyn aggregates in the ACC and hippocampus of oligomer-infused rats. Previous studies have reported the cortical spreading and cortical aggregates of p129-αSyn following intrastriatal infusion of αSyn fibrils or striatal overexpression of αSyn, indicating that the cortex is a final target of the spreading process [63,64,65,66]. The process of αSyn aggregation generates a variety of intermediate small structures that end up into insoluble fibrillar aggregates within LBs, in a self-fueling loop that origins and spreads multiple generations of toxic species [49,67,68]. Accordingly, small oligomeric and pre-fibrillar species are significantly abundant in the cerebrospinal fluid of PD patients [51,69]. In the present study, we suggest that soluble first- or second-generation intermediate species, resulting from the infusion of *in vitro* purified HαSynOs, may reach cortical regions and trigger cortical impairment. However, the fibrillar aggregates have not yet formed at the analyzed time point, and we suggest that a similar mechanism may apply to mild cognitive decline in the absence of cortical LB in PD patients with MCI [12]. 

### 4.2. The Intranigral Infusion of H-αSynOs Induced a Neuroinflammatory Response in Cognition-Related Brain Regions

A large body of literature points to neuroinflammation as a pivotal player in PD neuropathology [14]. Neuroinflammation has been widely described in the PD brain in postmortem studies, as well as in vivo, by PET imaging in PD patients [8,16,70,71,72,73], and has been largely investigated in preclinical models of PD as a critical neuropathological substrate of neurodegeneration in motor-related areas. In contrast, neuroinflammation has been poorly investigated in cognition-related regions and in relation to cognitive decline in PD [16]. However, a dysregulated immune response has often been associated with impaired cognition in other neurological conditions, suggesting that neuroinflammation might be investigated as a neuropathological correlate of cognitive decline in PD [74,75]. In the quest for an association between neuroinflammation and cognitive dysfunctions, some human imaging studies have revealed the presence of an inflammatory environment in limbic areas of PDD [15,76] and dementia with Lewy bodies patients [77].

A recent study conducted in the brain of PDD patients reported a correlation between microglial activation, T cell infiltration and α-Syn pathology in cognition-related brain regions, suggesting an α-Syn-driven neuroinflammatory response in PDD [8]. Interestingly, this was the only study that explored cytokine expression in cognitive brain regions, reporting an increase of IL-1β expression in the frontal cortex of PDD patients, yet without significant microgliosis [8]. In addition, the inter-relationship between α-Syn toxicity, neuroinflammation and cognitive decline has never been investigated in rodents, mainly due to the unavailability of a valid preclinical neuropathological PD model that recapitulates cognitive symptoms. In previous studies, we have shown that the intranigral infusion of H-αSynOs induced an intense neuroinflammatory response in motor areas including the SN and the nucleus striatum [27,29]. Here, we reported that the H-αSynOs infusion triggered a glial response in cortical regions typically involved in cognitive impairment in humans, including the ACC and hippocampus. Interestingly, alike the previous report on PD patients [8], we did not observe microglial proliferation in most of the analyzed subregions of the ACC and hippocampus, in line with the lack of neuronal loss in these regions. A small but significant increase of Iba-1 immunoreactivity was only observed in discrete hippocampal subregions, such as the CA1 pyramidal layer. While such heterogeneity is interesting in relation to the complex structure of the hippocampus and different roles of hippocampal subregions in memory processes, CA1 is one of the earliest sites within the hippocampus to be affected in PD-MCI patients [78,79]. The CA1 is also the most vulnerable subfield in AD, and CA1 microglia was more profoundly affected in AD [80]. Notably, the microglial production of the pro-inflammatory cytokine TNF-α was largely increased, suggesting that these cells acquired an altered phenotype in response to the infusion of H-αSynOs. Several studies have shown that aggregated and mutated αSyn stimulates pro-inflammatory responses in microglia, with soluble oligomer/protofibril forms holding a greater inflammatory potential [81,82] than the native monomeric protein [83,84]. Oligomeric αSyn triggers the inflammatory cascade in microglia via NF-kB activation subsequent to TLR2 binding [85,86]. An excess of pro-inflammatory cytokines, and specifically TNF-α, may play a critical role in the cortical dysfunction underlying memory deficits in our model. Besides its main role as a master regulator of inflammation, TNF-α is a key player in the regulation of neuronal excitability, synaptic strength and plasticity [26,87,88]. Of note, TNF-α is elevated in several neurological diseases associated with memory and learning deficits [89] and plays an important role in age-related cognitive decline [90] and in Alzheimer disease-induced cognitive impairment [91,92]. For instance, by binding to the neuronal TNF receptor TNFR1, TNF-α inhibits theta-burst-induced long-term potentiation (LTP) in CA1 synapses [93,94,95] and mediates amyloid-beta-induced inhibition of LTP in the DG [96]. Importantly, TNF-α may cause a reduction of synaptic strength via the direct or astrocyte-mediated regulation of AMPA receptor trafficking [21,88]. We therefore evaluated this TNF-α-related effect in the ACC and found a reduction in the expression of GluR1 in rats infused with H-αSynOs, further supporting the reduced neuronal activity in this area and suggesting a possible mechanism of inflammation-mediated cognitive decline. In line with the present interpretation, in a previous study, deficits in spatial learning induced by D-galactosamine administration caused an increase of the expression of proinflammatory cytokines in the hippocampus and a dysregulated glutamate receptor expression including the downregulation of the GluR1 AMPA subunit [97]. Notably, the differential expression of microglial TNF-α in the hippocampal subfields revealed a highest susceptibility of the Cornu Ammonis region in this animal model. Albeit such a differential inflammatory response has not yet been reported in PD-MCI patients, CA1/CA2/CA3 subfields represent the first hippocampal regions that undergo to atrophy in the conversion from PD-NCI (no cognitive impairment) to PD-MCI [78]. 

In conclusion, we present a neuropathological rat model that reproduces memory impairment featuring PD and may be relevant for the study of the cognitive dysfunction that accompanies the disease. In this model, the intranigral infusion of toxic oligomeric species of α-Syn drove a neuroinflammatory response with increased TNF-α production in distant cognition-relevant regions. We suggest that such oligomer-induced microglial dysfunction in the ACC and hippocampus may alter the neuronal activity via the action of TNF-α on neurons, leading to cognitive deficits.

## Figures and Tables

**Figure 1 cells-11-02628-f001:**
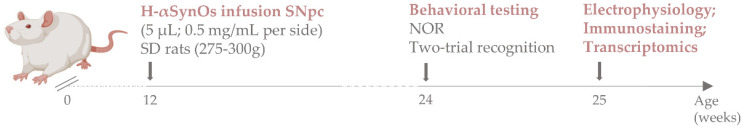
**Experimental workflow**. SD rats were infused intranigrally with H-αSynOs or PBS at three months of age. After three months, one group of rats underwent behavioural testing, followed by immunostaining and transcriptomic analysis. A separate group of rats was used for electrophysiology experiments. BioRender.com was used to compose this figure.

**Figure 2 cells-11-02628-f002:**
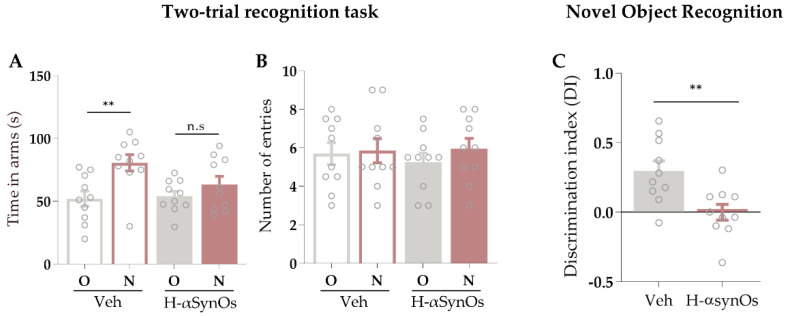
**Short-term memory was impaired following intranigral H-αSynOs infusion**. (**A**) Time spent (s) in the novel arm (N) versus the other arm (O) in the Two-trial recognition test in a Y maze by vehicle-infused and H-αSynOs-infused rats (n = 10; data are presented as the mean ± S.E.M; ** *p* < 0.01 by two-way ANOVA followed by Tuckey’s post hoc test). (**B**) Frequency of entrance in the N and O arms (n = 10; *p* > 0.05 by two-way ANOVA and Tuckey’s post hoc test). (**C**) Discrimination index measured by the novel object recognition test for vehicle-infused and H-αSynOs-infused rats (n = 10; ** *p* < 0.01 by Unpaired Student’s *t* test); n.s. = not significant.

**Figure 3 cells-11-02628-f003:**
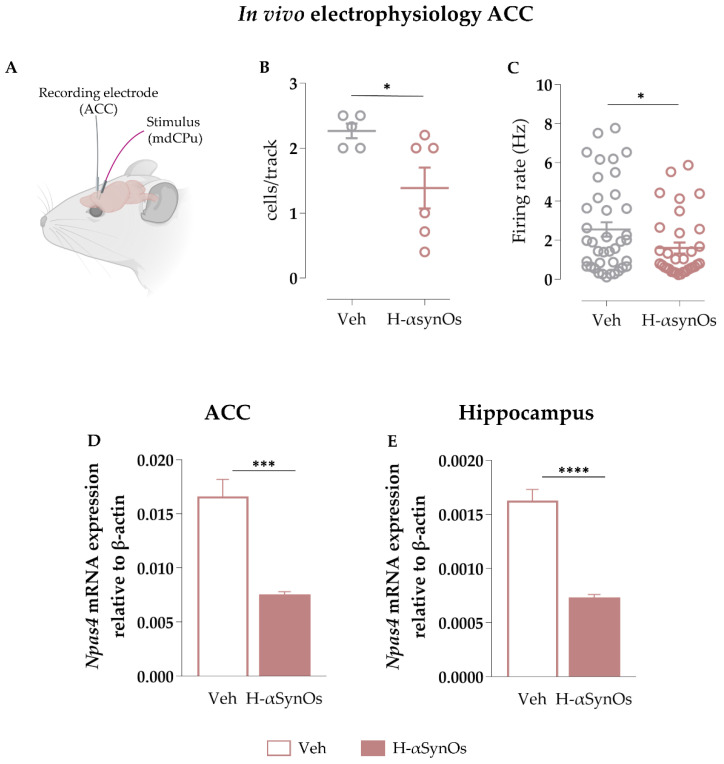
**Intranigral H-αSynOs infusion altered synaptic activity in limbic areas**. (**A**) Schematic representation of in vivo electrophysiological recordings in the ACC. (**B**) Mean number of spontaneously active cells recorded per track in the ACC. Single dots represent rat’s mean (n = 5–6; * *p* < 0.05 by Unpaired Student’s *t* test). (**C**) Mean firing frequency of pyramidal neurons. Dots represent single cells (n = 33–39; * *p* < 0.05 by Unpaired Student’s *t* test). (**D**,**E**) Npas4 mRNA expression levels measured by RT-qPCR, normalized to the housekeeping gene β-actin in the (**D**) ACC (n = 5; *** *p* < 0.001 by Unpaired Student’s *t* test) and (**E**) hippocampus (n = 5; **** *p* < 0.0001 by Unpaired Student’s *t* test). BioRender.com was used to compose the figure.

**Figure 4 cells-11-02628-f004:**
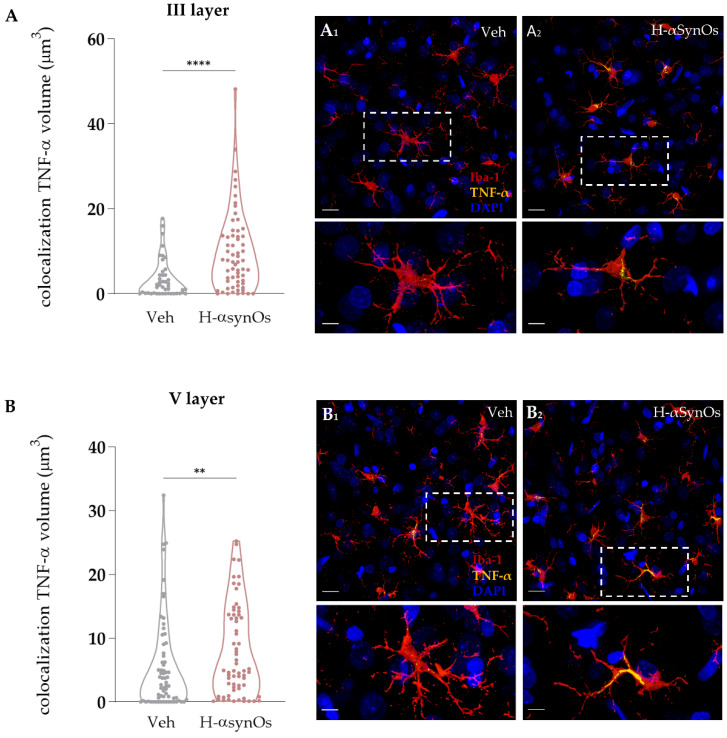
**Microglial TNF-α content was increased in the anterior cingulate cortex following H-αSynOs infusion**. Volume of TNF-α colocalized within microglial cells in the (**A**) III (n = 50–60 cells, n = 6 animals per group; **** *p* < 0.0001 by Mann–Whitney test) and (**B**) V cortical layer (n = 65 cells, n = 6 animals per group; ** *p* < 0.01 by Mann–Whitney test). Representative images of TNF-α (yellow) (**A_1_**,**A_2_**,**B_1_**,**B_2_**) colocalized with Iba-1^+^ cells (red). Magnification 63×. Scale bars: 20 μm; 5 μm.

**Figure 5 cells-11-02628-f005:**
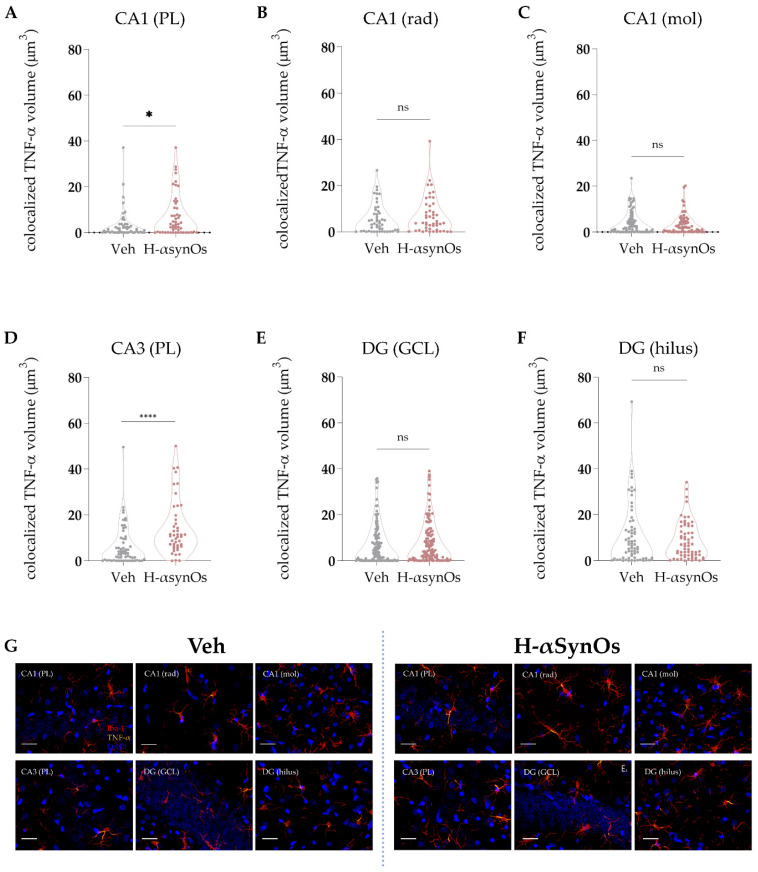
**Microglial TNF-α content was increased in the hippocampal CA1 and CA3 subregions following H-αSynOs infusion**. (**A**–**F**) Volume of colocalized TNF-α within Iba-1^+^ cells in discrete subfields of the dorsal hippocampus (n_(CA1-GCL)_ = 50−55, n = 6 animals per group; * *p* < 0.05 by Mann–Whitney test; n_(CA3-GCL)_ = 60−45, n = 6 animals per group; **** *p* < 0.0001 by Mann–Whitney test). (**G**) Representative images of TNF-α (yellow) colocalized within Iba-1^+^ cells (red). Magnification 63×. Scale bars: 20 μm. GCL (granular cell layer); rad (radiatum); mol (molecular); DG (dentate gyrus), PL (pyramidal layer); n.s. = not significant.

**Figure 6 cells-11-02628-f006:**
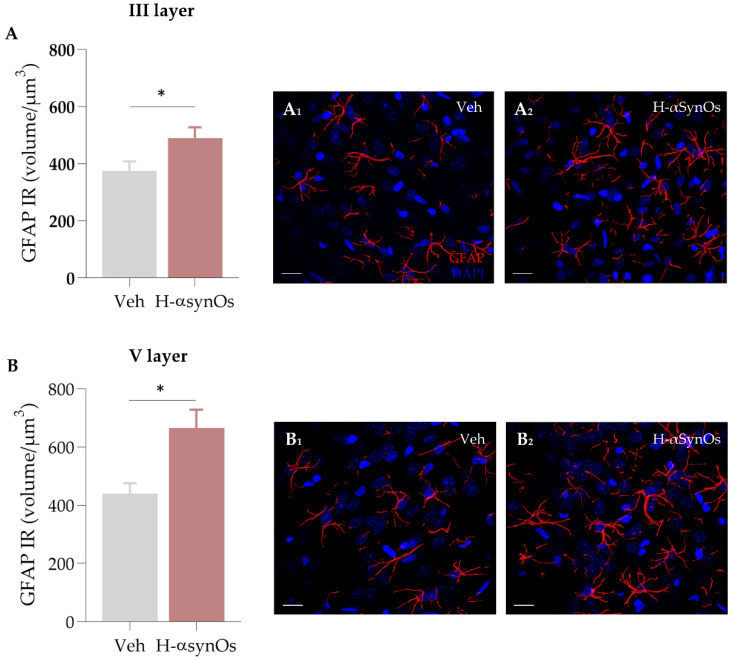
**GFAP immunoreactivity in the anterior cingulate cortex following after H-αSynOs infusion**. Total volume occupied by GFAP^+^ cells in the cingulate cortical layers (**A**) III (n = 80–110 cells, n = 6 animals per group; * *p* < 0.05 by Mann–Whitney test) and (**B**) V (n = 90 cells, n = 6 animals per group; * *p* < 0.05 by Mann–Whitney test). Representative images of GFAP (red) in the (**A_1_**,**A_2_**) III and (**B_1_**,**B_2_**) V layer of ACC. Magnification 63×. Scale bar: 20 μm.

**Figure 7 cells-11-02628-f007:**
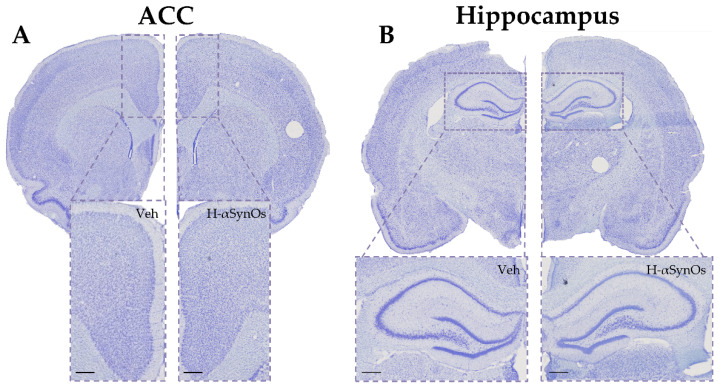
**Intranigral H-αSynOs infusion did not cause neuronal cell loss in limbic areas**. Representative images of Nissl-stained sections of (**A**) ACC and (**B**) dorsal hippocampus. Higher magnification pictures are shown (100 μm) below each image.

**Figure 8 cells-11-02628-f008:**
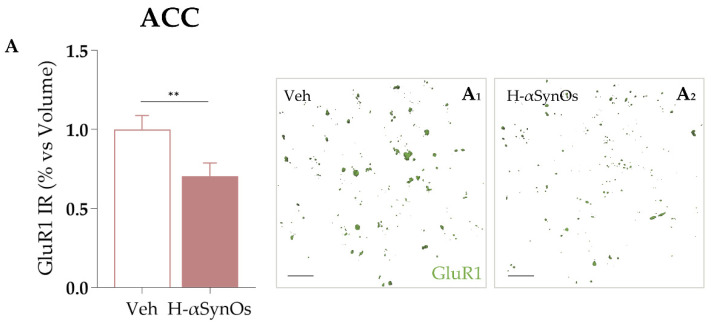
**Cortical GluR1 expression is decreased following H-αSynOs oligomers infusion**. (**A**) GluR1 expression levels in the ACC (n = 50 subfields, n = 6 animals per group; ** *p* < 0.01 by Mann–Whitney test). Representative images of GluR1 (green) expression in the ACC of (**A_1_**) vehicle-infused and (**A_2_**) H-αSynOs-infused rats. Magnification 63×. Scale bar: 20 μm.

**Figure 9 cells-11-02628-f009:**
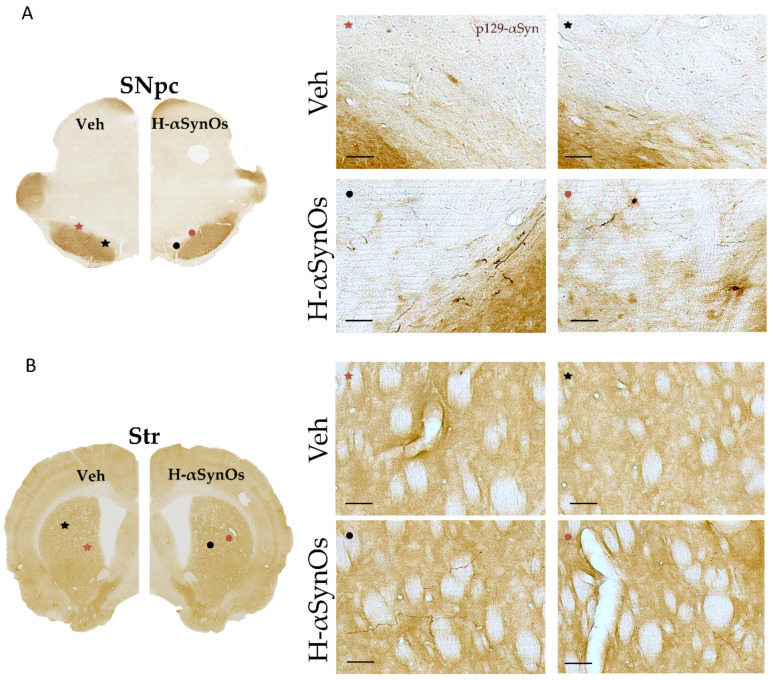
**Intranigral infusion of H-αSynOs increases p129-αSyn aggregates in the substantia nigra (A) and striatum (B).** Representative pictures showing aggregates of p129-αSyn in the substantia nigra pars compacta (SNpc, (**A**)) and striatum (Str, (**B**)). Higher magnification pictures are shown on right panels (scale bar: 50 μm).

**Table 1 cells-11-02628-t001:** Iba-1^+^ immunoreactivity in anterior cingulate cortex and dorsal hippocampus following H-αSynOs intranigral infusion.

**ACC (Iba-1 IR; Volume/mm^3^)**			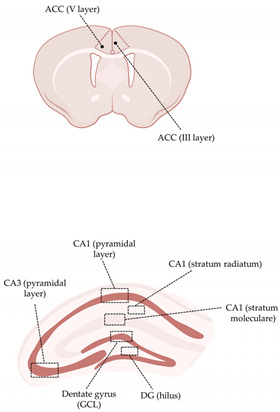
	**Veh**	**H-αSynOs**	
*III layer*	318.4 ± 23.02	357.1 ± 25.25	
*V layer*	348.6 ± 24.80	380.2 ± 26.42	

**Dorsal Hippocampus (Iba-1 IR;** **Volume/mm^3^)**			
	**Veh**	**H-αSynOs**	
*DG (GCL)*	335.2 ± 16.68	348.6 ± 22.24	
*DG (hilus)*	320.8 ± 16.53	387.2 ± 20.30 *	
*CA3 (pyramidal layer)*	280.8 ± 18.75	249.7 ± 16.52	
*CA1 (pyramidal layer)*	463.1 ± 28.13	600.5 ± 30.42 **	
*CA1 (stratum radiatum)*	564.2 ± 28.49	644.5 ± 39.35	
*CA1 (stratum moleculare)*	899.7 ± 40.77	843.6 ± 44.55	

Each reported value represents the total volume occupied by Iba-1^+^ cells in the selected subregions. Values represent the mean ± SEM. * *p* < 0.05; ** *p* < 0.01 by Mann–Whitney test. BioRender.com was used to compose the figure.

## Data Availability

Not applicable.

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
