# Peer review of "The Intranigral Infusion of Human-Alpha Synuclein Oligomers Induces a Cognitive Impairment in Rats Associated with Changes in Neuronal Firing and Neuroinflammation in the Anterior Cingulate Cortex"

_cells, 2022, doi:10.3390/cells11172628_

Round 1

Reviewer 1 Report

The goal of the study is to demonstrate the feasibility of the animal model of PD to show cognitive deficits using standard behavioral paradigms.

This manuscript describes  correlation studies between intracerebral infusion of pre-formed human alpha synuclein oligomers (H-αSynOs) within the substantia nigra pars compacta ,cognitive deficit, neuroinflammation and neural deficit in ACC in rodents.

The experiment were very well described and conducted using published protocols for surgery, behavior, molecular biology and histology techniques.

The following are points to be revised:

·         The  increase of Iba-1 in CA1-pyramidal layer after H-αSynOs  should be discussed.

·         163 line typo. Experimental workflow. SD also this section should be include in the animal surgery

·         Correct : CA1 and CA3 has not granular layer, replace by pyramidal layer; same for stratum radiens= stratum radiatum

·         Figure 1 more specific details of the interventions. Clarify if the injection of oligomer was day 0 or 12, add 1st behavior study (week 12) in the graphic design.

·         Clarify / add information about control of the oligomer injection, volume etc.

·         More details of the co-localization analysis is needed.

·         Although the number of entries in the arms did not differ across experimental groups, a locomotor behavioral analysis is needed to determine any bias in cognitive paradigms which are depend of locomotion.

·         Explain why  H-αSynOs  induces ACC neuronal deficits and CA1 increase of microglial marker

Author Response

We highly appreciate and thank the reviewers for the positive comments to our manuscript as well as the suggestions to further improve it, which have been addressed as detailed below:

Q1: The increase of Iba-1 in CA1-pyramidal layer after H-αSynOs should be discussed.
A1: We have now implemented this part in the discussion (lines 579-584).

Q2: 163 line typo. Experimental workflow. SD also this section should be included in the animal surgery.
A2: Thank you for the observation, this part has been emended accordingly to reviewer comment.

Q3: Correct: CA1 and CA3 has not granular layer, replace by pyramidal layer; same for stratum radiens= stratum radiatum
A3: Thank you, all corrections have been made as requested.

Q4: Figure 1 more specific details of the interventions. Clarify if the injection of oligomer was day 0 or 12, add 1st behavior study (week 12) in the graphic design.
A4: We agree that figure 1 was misleading and we corrected it accordingly. We also clarified that the numbers indicate the age of animals in weeks.

Q5: Clarify / add information about control of the oligomer injection, volume etc.
A5: Thank you, this information has been added in the method section (lines 170-171).

Q6: More details of the co-localization analysis is needed.
A6: The description of the colocalization analysis has been extended (lines 268-275). 

Q7: Although the number of entries in the arms did not differ across experimental groups, a locomotor behavioral analysis is needed to determine any bias in cognitive paradigms which are dependent of locomotion.
A7: We thank the reviewer for this comment. The Y maze is structurally not a suited means to measure motor activity. Indeed, the opacity of the maze walls impedes crossing of infrared beams, thus not allowing the use of automated motor evaluation counters based in emitter-detector systems, like the ones we have in our laboratories. In previous studies we have performed the open field test, which did not reveal any signs of hypokinesia in oligomer-infused rats. Instead, we routinely perform in this PD model the more subtle challenging beam walking test, which reveals the presence of sensorimotor impairment, to assess the symptomatic correlates of neurodegeneration. We have now added a graph in supplementary material showing the results of such test. Specifically, the test shows that rats display a sensorimotor impairment, but the time spent to cross the beam was similar for parkinsonian and control rats, suggesting that parkinsonian rats do not have any major hypokinetic deficit, in accordance with the moderate extent of nigral neurodegeneration obtained with this protocol (see also Boi et al, 2020 for more details). This part has been implemented in the results section (lines 360-362).

Q8: Explain why H-αSynOs induces ACC neuronal deficits and CA1 increase of microglial marker.
A8: We expanded the interpretation of our results in the discussion (lines 617-621). CA1 increase of microglial marker was further discussed in discussion (lines 579-584).

Reviewer 2 Report

I read with interest the article by Palmas and colleagues, which studied the effect of bilateral intranigral infusion of oligomeric species of alpha-synuclein on cognitive function. The article is well written and analyzes an important issue. 

I have, however, some concerns: 

  • Did the authors confirm the presence of oligomers? If yes, how?
  • Why did the authors use human alpha-synuclein oligomers to induce pathology? The nucleation reaction can be different if used alpha-synuclein from rats or mice, and it can be more efficient even in the generation of oligomeric species. 
  • The concentration of oligomers is very high. I’m aware this is the general standard; however, this is not what happens physiologically in PD subjects. Every protein, even in oligomeric conformation or misfolded, can have a different function based on the relative levels they reach in fluids where they carry their function. If high concentration can be used in some experiments, I don’t think that high levels can give information on physiological function.
  • It is not clear to me if their models also have some motor features as shown in previous publications. If not, how do the authors explain it? If yes, how did the author evaluate them?
  • The title of paragraph 3.2 is “H-αsynOs infusion altered neuronal activity in the anterior cingulate cortex and hippocampus;” however, I don’t see the results of the hippocampus measurements, but only the expression of Npas4. Or do the authors with pyramidal neurons also refer to the ones in the hippocampus? If yes, it should be added to the method section because, at the moment, in the method for pyramidal neurons they refer only to the ones in the ACC (moreover, the title of paragraph 2.8 is “In vivo single-unit recordings from the ACC” with no mention of the hippocampus). Or do the authors for neuronal activity in the hippocampus refer only to Npsa4 measures? If yes, why did the author exclude the hippocampus or other important areas?
  • It would be nice to add the list of the other genes altered and a better explanation of why they focused only on Npas4. Moreover, adding the levels of protein would be preferable since it is known that the IEGs often do not correlate with protein levels.

Minor point:

  • In figure 1 it looks like the infusion was done at 12 weeks but in the text looks like it was done at 24 weeks.

Author Response

We highly appreciate and thank the reviewers for the positive comments to our manuscript as well as the suggestions to further improve it, which have been addressed as detailed below:

Q1: Did the authors confirm the presence of oligomers? If yes, how?
A1: In the original study by Boi et al 2020, we used fluorescent oligomers to confirm the specificity of infusion site and to determine the appropriate oligomer dose/concentration. Here, we used the same infusion protocol, and we determined the oligomer-induced spreading of endogenous alphaSyn by analysing the presence of endogenous phospho-alphaSyn, derived from the seeding and spreading of infused oligomers. We could nicely describe presence of phospho-alphaSyn in the infusion site (substantia nigra) and its projection area striatum by DAB immunohistochemistry. Phospho-alphaSyn immunoreactivity was also observed in other anatomically connected areas such as the subthalamic nucleus (data not shown). However, besides very small immunoreactive dots, we did not detect clear aggregates of phospho-alphaSyn in the ACC neither in the hippocampus (lines 463-465).

Q2: Why did the authors use human alpha-synuclein oligomers to induce pathology? The nucleation reaction can be different if used alpha-synuclein from rats or mice, and it can be more efficient even in the generation of oligomeric species.
A2: We do agree with the reviewer that alpha-synuclein from rats or mice may induce some different nucleation. However, we have shown in previous papers and in the present one that the human alpha synuclein oligomers induce spreading from the injection site to anatomically distant regions (Boi et al, 2020).
There are several reasons for originally choosing these oligomers instead of other available forms such as mixed prefibril/fibrils. We exploited a protocol set by Prof. Alfonso de Simone, which enables to prepare and purify homogeneous oligomers in relation to both size and tri-dimensional structure. While the oligomeric species are believed to play a major role in PD neuropathology, in previous outstanding studies Prof. De Simone characterized the most neurotoxic oligomeric species against neurons (Fusco et al, Science 2017; 358: 1440–1443). We therefore decided to characterize the toxicity of these size/structural specific species in vivo. A main advantage of using this preparation is the infusion of a highly homogeneous solution of these specific oligomeric species, in contrast with mixed species used in other studies, which allow a high reproducibility of the neuropathology across animals and across experiments.

Q3: The concentration of oligomers is very high. I’m aware this is the general standard; however, this is not what happens physiologically in PD subjects. Every protein, even in oligomeric conformation or misfolded, can have a different function based on the relative levels they reach in fluids where they carry their function. If high concentration can be used in some experiments, I don’t think that high levels can give information on physiological function.
A3: In the original study showing the toxicity of these oligomeric species for the first time, we conducted a dose-response preliminary experiment to establish the optimal dose/concentration to achieve neurodegeneration without mechanically damage the infusion area (Boi et al, 2020). We do agree that this dose is not appropriate to investigate physiological functions of alpha synuclein, nor to investigate any possible, although unlikely, physiological function of oligomers.  However, the aim of the present study is to investigate the neuropathological role of these specific toxic oligomeric species. It is worth to notice that oligomers concentration has been found elevated in biological fluids of parkinsonian patients. We have now implemented this information in the discussion (lines 488-491).

Q4: It is not clear to me if their models also have some motor features as shown in previous publications. If not, how do the authors explain it? If yes, how did the author evaluate them?
A4: We thank the reviewer for this important observation. As specified in the results section (lines 360-362) we have now added the results of a sensorimotor test which we routinely perform in this model to confirm the presence of motor symptomatology (see supp. Material). With the challenging beam walk test we evaluated specifically motor coordination and balance, which resulted to be significantly affected in our model at this specific time point.

Q5: The title of paragraph 3.2 is “H-αsynOs infusion altered neuronal activity in the anterior cingulate cortex and hippocampus;” however, I don’t see the results of the hippocampus measurements, but only the expression of Npas4. Or do the authors with pyramidal neurons also refer to the ones in the hippocampus? If yes, it should be added to the method section because, at the moment, in the method for pyramidal neurons they refer only to the ones in the ACC (moreover, the title of paragraph 2.8 is “In vivo single-unit recordings from the ACC” with no mention of the hippocampus). Or do the authors for neuronal activity in the hippocampus refer only to Npsa4 measures? If yes, why did the author exclude the hippocampus or other important areas?
A5. We agree that the title of the paragraph might sound equivocal. We therefore split the results in two paragraphs. Moreover, we would like to specify that the in vivo electrophysiological recordings were made only in the ACC for pure technical reasons. In fact, in vivo electrophysiological recording of the hippocampus is usually not recommended given the high structural complexity of this area, which does not allow to discriminate with sufficient confidence the cells of interest. For that reason, we decided to record the neuronal activity in the ACC, and to confirm changes in neuronal activity by using a classical IEG analysis.

Q6: It would be nice to add the list of the other genes altered and a better explanation of why they focused only on Npas4. Moreover, adding the levels of protein would be preferable since it is known that the IEGs often do not correlate with protein levels.
A6: The bulk sequencing experiments revealed 400 and 300 differentially expressed genes respectively in the ACC and hippocampus. We then performed a clusterization of those genes differentially expressed both in the ACC and hippocampus based on their biological function. From this analysis we identified about 10 genes involved in the regulation of neuronal activity. Among them, Npas4 was particularly intriguing since this gene is neuron-specific and plays a role in the modulation of neuronal activity in relation to memory formation. Following the reviewer suggestion, we have now implemented this part in the discussion (lines 520-525). Given the complexity of such results, also in terms of their multiple meanings, we are currently preparing a manuscript fully dedicated to transcriptomics.
IEGs analysis is a classical means commonly used to assess neuronal function in models of PD, because their almost instantaneous activation allows to identify the neuronal population directly involved in activity changes. For all these reasons we think it is a good marker of altered neuronal activity in the present study.